# Large-scale ion generation for precipitation of atmospheric aerosols

**Shaoxiang Ma [(a)1], He Cheng[(a)1], Jiacheng Li[1], Maoyuan Xu[1], Dawei Liu[1], and Kostya (Ken) Ostrikov[2]**

[1]State Key Lab of Advanced Electromagnetic Engineering and Technology, School of Electronic and Electrical Engineering, Huazhong University of Science and Technology, WuHan, HuBei 430074, China

[2]Institute for Future Environments and School of Chemistry and Physics, Queensland University of Technology, Brisbane, Queensland 4000, Australia

[a] Equal contribution

**Correspondence:** Dawei Liu (liudw@hust.edu.cn)

**Abstract.** Artificial rain is explored as a remedy to climate change caused farmland drought and bushfires. Increasing the ion density in the open air is an efficient way to generate charged nuclei from atmospheric aerosols and induce precipitation or eliminate fog. Here we report on the development of the large commercial installation scale atmospheric ion generator based on corona plasma discharges, experimental monitoring and numerical modeling of the parameters and range of the atmospheric ions, and application of the generated ions to produce charged aerosols and induce precipitation at a scale of a large cloud chamber. The coverage area of the ions generated by the large corona discharge installation with the 7.2 km long wire electrode and applied voltage of -90 kV is studied under prevailing weather conditions including wind direction and speed. By synergizing over 300,000 localized corona discharge points, we demonstrate a substantial decrease of the decay of ions compared to a single corona discharge point in the open air, leading to a large-scale (30 m×23 m×90 m) ion coverage. Once aerosols combine with the generated ions, charged nuclei are produced.

The higher   wind speed has led to the larger areas covered by the plasma generated ions. The cloud chamber experiments (relative humidity $130\pm10\%$) suggest that the charged aerosols generated by ions with the density of $\sim 10^4/cm^3$ can accelerate the settlement of moisture by 38%. These results are promising for the development of large-scale installations for the effective localized control of atmospheric phenomena.

## 1. Introduction

The water cycle and rainfall on the Earth are affected by the climate change, causing widespread droughts around the world (Dai, 2013; Trenberth et al., 2014). Recently, more and more forest fires happened in North America, Australia and China. Besides that, the lack of water in many parts of Asia and Africa seriously influence the local agriculture, industry, and human health (Jolly et al., 2015; Lesk et al., 2016). The artificial rain has been used widely in many countries to alleviate the drought problems by enhancing precipitation. The artificial rain is commonly realized by dispersing substances, such as silver iodide, dry ice and table salts, which act as cloud condensation nuclei and alter the physico-chemical processes within the cloud.

It is well known that the ions generated by galactic cosmic rays in the atmosphere directly affect the changes in cloudiness of the Earth (Carslaw et al., 2002; Pierce and Adams, 2009). The experiment in the Wilson chamber suggested that ions generated by radioactive materials acted as condensation nuclei under supersaturation conditions (Yang et al., 2018). Nielsen *et al.* found that the charged nuclei could remain in the condensation phase even when the relative humidity is less than 100% (Nielsen et al., 2011). This is why plasma- and laser-based techniques have been employed to generate the charged nuclei in the open air to try to realize rain enhancement or, alternatively, fog elimination (Henin et al., 2009; Khain et al., 2004; Tan et al., 2016).

Ideally, the effects (e.g., precipitation of atmospheric aerosols) of the plasma-based methods should be delivered over the areas of small-to-medium-size farmlands. Indeed, it appears appealing to install, for example, a large-scale corona plasma discharge system on top of a hill and distribute the generated ions using the wind. Indeed, corona plasma discharges can release high-concentration ion fluxes, which can not only be transported to a large area by the upwind slope airflow, but also effectively interact with atmospheric aerosols thereby generating charged nuclei needed to induce or enhance water precipitation in the areas along the downwind direction. However, there are formidable science and technological challenges to implement it in practice. First, we are not aware of any systematic experimental and theoretical studies on the generation efficiency of ions, the

coverage area of ions and effect of wind on the ion transport at the relevant scale of the atmospheric ion generating installations. Second, the high efficiency of the ion generation in the laboratory does not necessarily translate into a similar performance at a scale. One reason is destructive interference of multiple corona discharges along the same plasma-generating wire, addressed in this work. Third, there are no reliable estimates of the volumes where the ions are generated at concentrations that are appropriate for atmospheric aerosol precipitation. Fourth, the reliable quantitative estimates of the significance of the effects of plasma-generated ions on moisture precipitation under real-world conditions are very limited, if available at all. Fifth, quantitative effects of the real-world wind on the ion coverage at the large installation scale are eagerly anticipated by the community.

In this work, we address all the above issues and report on the coverage area of ions generated by a large corona discharge system with the 7.2 km long wire electrode and applied voltage of -90 kV. The vertical and horizontal distribution of the ions in the downwind are measured under realistic weather conditions. The numerical models of the corona discharge and ion transport are developed. The results of numerical modelling are consistent with the experiment results, and indicates that the ions produced    by the plasma discharge device may affect aerosol precipitation over large areas depending on the direction and speed of the wind. The effects of the plasma generated ions on moisture precipitation are studied experimentally at a scale of a large cloud chamber.

## 2. Methods

### 2.1 Corona discharge installation and ion measurements

Both the positive and negative corona discharge can be used to increase the ion density in the open air. Under the similar conditions of the electric circuit the loss of the positive corona is greater than that of negative corona at the same applied voltage. Because the negative corona curve is flatter and since larger negative corona currents can be obtained, the negative corona is much better adapted for the application such as fog elimination and electrical precipitation than the positive corona.(Sawant et al., 2012; Strong, 1913)

The wire electrode is a low cost and high efficiency plasma source configuration, especially for the

large scale corona discharge system. For the wire electrode radius within the range of 100 μm to

1000 μm, the plasma thickness increases with increasing wire radius. The larger wires can generate

more electrons, however, the electron energy decreases due to the lower electric field near the larger

wire(Chen and Davidson, 2003). Stainless steel stranded wire is suitable wire electrode material

considering durability and stability.

Electrons generated by negative corona discharge attach to electronegative gas molecules (such as,

$O_2$) to generate negative ions ($O_2^-$). Recombination of electrons with positive ions is negligible.

Therefore, ionization competes primarily with electron attachment. The ionization predominates over

the electron attachment and new electrons are generated. The rate of ionization balances the rate of

electron attachment at the reduced electric field of 120 Td (1 Td $=10^{-21}$ Vm$^2$). Beyond this ionization

boundary, the attachment dominates over the ionization, and the electron density decrease gradually

as the electric field decreases.(Chen and Davidson, 2003; Kossyi et al., 1992; Lowke and Morrow,

1994) In the region away from the electrode, because the absence of the electric field, the charged

particles, including electrons and ions, perform a faster decay through the electron-ion and ion-ion

recombination with background charged particles.(Xiong et al., 2010)

The large corona discharge system employed in this study has the floor area ~11304 m$^2$ (**Figure 1** (a)

and (b)). Six poles each with the height of 20 m supporting the 7.2 km long wire electrode are

erected vertically and arranged in a regular hexagon array. The wire electrodes are divided into two

layers, at the height of 20 m and 15 m, i.e., with 5 m separation between the layers. There are 10 wire

electrodes in each layer, and the horizontal distance between the wires is 50 cm to avoid the mutual

interference. The stainless-steel wire has six strands and a diameter of 1 mm. The high voltage DC

power supply is Technix 44-2015, which can monitor the output voltage and current value in real

time.

**Figure 1** (c) shows the satellite image of test zone. The hexagon shows the large-scale corona

discharge system. The red points show the locations of the ion density measurements. The hydrogen

balloon carrying the ion counter (Air Ion Counter) is used to measure the vertical (5 m – 50 m) and horizontal (20 m – 50 m) ion density in the downwind as shown in Fig. 1(d). The horizontal distance between the hydrogen balloon measurement and the wire electrode is 20 m to ensure the safety of the experiment. The image of the corona discharge on the wire electrode is taken by digital camera Nikon D800 with the exposure time of 2 s. The optical emission spectroscopy (OES) of the negative corona discharge on the wire electrode with the applied voltage of -40 kV is measured by an optical spectrometer (Ocean optics USB4000+).

## 2.2 Ion transport model

The 2D numerical model is used to the study the distribution of ions within 1 m from the wire electrode. The model used in this study extends the existing models (He et al., 2013; Liu et al., 2012, 2014a, 2014b) to model the relevant phenomena at the relevant scales. The model solves the Poisson's equation and the transport equations for neutral and charged species as a function of time. The number density of each species is obtained by solving the continuity equation. The electric field is obtained by solving the Poisson's equation. The electron energy is calculated by the electron energy conservation equation.

The transient advection diffusion reaction equation (equation (1)) is used to study the effect of wind on the transport of ions (Albani et al., 2015; Ashrafi et al., 2017; Schleder and Martins, 2016)

$$\frac{\partial c}{\partial t} + u\nabla c = \nabla(K\nabla c) - \lambda c, \ c = c(x,y,z,t) \qquad , \qquad (1)$$

where $c$ is the crosswind-integrated concentration of ions, $t$ is time, $u$ is the wind speed, $\nabla = (\frac{\partial}{\partial x}, \frac{\partial}{\partial y}, \frac{\partial}{\partial z})$ , x is horizontal downwind direction, y is horizontal radial direction, z is vertical direction, $K$ is the eddy diffusivity, which calculated the meteorological data of the test place (~4.82 m$^2$/s)(Albani et al., 2015), $\lambda$ is the decay constant, which calculated based on the recombination of positive ions and electrons (~1.5113/s)(Sakiyama et al., 2012).

The eddy diffusion $K$ represents the diffusion of ions under the influence of the turbulent state of

atmosphere. During stable conditions, the maximum value of eddy diffusivity decreases with increasing stability. In stable conditions, a height at which turbulence maintained is limited by the destruction of turbulent kinetic energy by negative buoyancy(Ulke, 2000), while in unstable conditions, the maximum value of eddy diffusivity increases with growing instability characterized by increasing values of $H_A/L$ ($H_A$ is the ABL-height, L is the Monin-Obukhov length).

The decay constant $\lambda$, it represents the decay of ions due to the recombination reactions between charged particles, such as $e+N_2^+\rightarrow N_2$, $e+O_2^+\rightarrow 2O$, $O_2^-+N_2^++N_2\rightarrow 2O+2N_2$, etc. According to our simulation results, the combination of numerous corona discharge points actually decreases the decay of ions generated by a single corona discharge point in the open air.

## 2.3 Cloud chamber experiments

The cloud chamber used to study the enhanced water condensation by ions has the size of 3 m×1.5 m×1.5 m (**Figure 2**). The temperature inside the chamber is 2±1℃ during the experiment (Testo 605-H1). The dehumidifier (Gree DH40EF) and ultrasonic humidifier (Midea SC4E40) are used to control the humidity supersaturation at 130±10%. The laser illuminator (YGL01, planar laser source) is used to light up the cloud chamber. The changing process of droplets is recorded by the digital camera (Nikon D800) with a microscopic lens. Finally, an acceptor with the cross section of 10 cm$^2$ is placed at the bottom of the cloud chamber to accept the settlement of moisture.

## 3. Results and discussion

## 3.1 Corona discharges and ion generation

The applied voltage on the wire electrode with the length of 7.2 km is -90 kV (**Figure 1**), and the discharge current is 0.3 mA, so the plasma power is 27 W. **Figure 3** (b) shows that there are more than 20 corona discharge points on the wire electrode with the length of 1 m. Because the distance between the camera and the wire electrode is only 2 m, the applied voltage is therefore -40 kV to

ensure the experimental safety. Although the six strands of stranded stainless-steel wire have good anti-stretching   and anti-bending properties, there are numerous small peaks on the wires, which help ignite corona discharges. Consequently, the minimum inception voltage of the corona discharge on the wire is only -15 kV. More corona discharge points are thus expected for the field experiment with the applied voltage of -90 kV.

**Figure 3** shows the OES data of the corona discharge on the wire electrode. The brightest peaks can be seen around 315, 337, 350 and 380 nm, which are attributed to the nitrogen 2nd positive electronic transition ($N_2(C^3\Pi_u - B^3\Pi_g)$) and its family of vibrational rotational level sub-transitions (Antao et al., 2009; Hu et al., 2013; Wang et al., 2013; Zou et al., 2019). The OES of OH and O radicals are quite low compared with $N_2$ species( Gan et al., 2019).

For the negative corona discharge, the plasma region only occupies a very small portion of the space around the power electrode, negative ions occupy the region outside the plasma region (Chen and Davidson, 2003; Riba et al., 2018; Yao et al., 2019; Zhang et al., 2019). **Figure 4** (a) shows the boundary of the plasma region defined as the position where the reduced electric field (electric field divided by neutral density, E/N) of 80 Td (1 Td = $10^{-21}$ Vm$^2$ ) is 0.1 cm from the electrode (Chen and Davidson, 2003). Therefore, the negative ion density at the boundary of plasma region (~1cm from the electrode, reduced electric field ~10 Td) is chosen as the source density of negative ions for each case.

As the applied voltage amplitude increases from 30 kV to 90 kV, the source density at 1 cm from the single discharge point increases from $1.1\times10^8$/cm$^3$ to $3.9\times10^8$/cm$^3$, and the density at 1 m from the single discharge point increased from $1.8\times10^6$/cm$^3$ to $8\times10^6$/cm$^3$ (**Figure 4** (b)). In addition, **Figure 4** (c) suggests that the ion density in the whole simulation domain increases with the increasing applied voltage amplitude.

**3.2 Ion transport**

**Figure 5** shows the effect of the wind on the transport of ions generated by a single corona discharge

point. The ion density at the plasma boundary is chosen as the source density. The ion density measured at the position at 1 m from the electrode is consistent with the numerical result calculated by Eq. (1). As the wind speed increases from 0 to 5.7 m/s, the ion density at 1 m increases from $2.5 \times 10^6/cm^3$ to $4 \times 10^6/cm^3$, and the density at 3 m increases from $3.2 \times 10^4/cm^3$ to $2 \times 10^5/cm^3$, which indicates that the diffusivity dominates the ions movement in the low electric field region, and the wind convection can substantially enhance the ion transport (Albani and Albani, 2019; Ashrafi et al., 2017; Hosseini and Stockie, 2016; Schleder and Martins, 2016).

**Figure 6** (a) shows the coverage range (horizontal/vertical (x/z) direction) of negative ions generated by the large corona discharge system shown in **Figure 1**. In order to ensure the safety, the ion counter carried by the balloon starts recording the negative ion density at the position of 20 m from the wire electrode. The ion density decreases from $5 \times 10^5/cm^3$ to $8 \times 10^3/cm^3$ as the distance from the wire electrode increases from 20m to 30m. Afterwards, the ion density decreases to the background value as the horizontal distance increases to 40 m. The relatively high density $10^4/cm^3$ at the horizontal position of 50 m can be attributed to the random enhanced transport by gusts. The width of the coverage area in radial direction was 90 m. Therefore, the whole coverage volume was approximately 30 m (long) ×23 m (height) ×90 m (width). The overall discharge system configuration is an hexagon with side length of 60m (**Figure 1**), therefore, the distance between two opposite side is 103.92m. The radial measurement range is ~95m with the consideration of surrounding buildings. Therefore, the width of 90m is obtained. Because 90m is less than 103.92m, the decays at the boundary is avoided in this range. The coverage volume of 30m×23m×90m is a relatively conservative range. Aerosols can combine with the ions generated in this volume, thus creating a large number of charged nuclei within the volume (Anon, 2000; Bricard et al., 1968; Harrison, 2000; Harrison and Carslaw, 2003; Keefe et al., 1959).

It is interesting to note that the ion density at 20 m from the single corona discharge point is only $10^2/cm^3$ (**Figure 6** (c)), which is only 0.02% of the ion density generated by the large corona discharge system (**Figure 6** (a)). **Figure 3** (b) suggests the presence of more than 45 corona discharge points along the 1 m long wire electrode when the applied voltage is -40 kV, and the

average distance between corona discharge points is only 2.5 cm. For the large corona discharge system with the 7.2 km long wire electrode and applied voltage of -90 kV, the number of the corona discharge points is expected to be at least $3.2 \times 10^5$ ($45 \times 7.2 \times 10^3$).

**Figure 6** (d) shows the collective efficiency of ion transport of ions generated by the many corona discharge points. Because the distance between the corona discharge points is in the cm range and the coverage area of each corona discharge point is 25 m (x-direction) × 30m (y-direction) (**Figure 6** (c)), the overlap of the coverage areas results in the high ion density in the open air.

The comparison between the experimental (**Figure 6** (a)) and simulation (**Figure 6** (c)) results suggest that the combination of numerous corona discharge points actually decreases the decay of ions generated by a single corona discharge point in the open air, and substantially increases the outward transport capacity of ions of the large corona discharge system. In the simulation model, by decreasing the decay constant $\lambda$ by 4.533 times, the equation (1) provides a similar coverage area as measured in the experiment (compare **Figure 6** (a) and (b)).

Because the large corona discharge system may be setup on the mountain top, e.g., with an altitude of around 4,000 m (Farr and Chadwick, 2012; Mu et al., 2007; Shiyin et al., 2003; Zhou and Yang, 2010), the effect of the wind and applied voltage on the coverage area of large corona discharge system is also studied by the numerical model.

**Figure 7** (a) – (c) suggest that as the wind speed increases from 2.89 m/s to 12.5 m/s, the axial length of the coverage area (ion density $\sim 10^4/cm^3$) increases from 26 m to 73 m. **Figure 7** (d) – (f) suggests that as the amplitude of the applied voltage increases from 60 kV to 180 kV, the axial length of the coverage area increases from 34 m to 46 m. The comparison between the effects of the wind and applied voltage indicates that the increasing transfer flux with the faster wind speed is a more efficient way to increase the coverage area of the large corona discharge system. The larger coverage area can facilitate the generation of charged nuclei, which eventually realize rain enhancement and fog elimination (Frederick and Tinsley, 2018; Tinsley and Zhou, 2015; Zhang et al., 2018a).

## 3.3 Ion-enhanced precipitation

The cloud chamber is used to study the enhancement of droplets growth by ions before the completion of the large-scale corona discharge system.   The corona discharge on the needle electrode (applied voltage of -23 kV) provides an environment with the ion density of $1.2 \times 10^5/cm^3 \sim 2 \times 10^4/cm^3$ in the cloud chamber, which corresponds to the ion density in the region $30 - 35$ m from the wire electrode (**Figure 6** (a)). The high ion density in the cloud chamber facilitates the aerosols charging. Aerosols (diameter $> 0.7\mu m$) are mainly charged by the drift of ions on the electric field lines intersecting the surface of aerosols. The diffusion of ions to the surface of aerosols also contributes to the aerosols charging(Jidenko and Borra, 2012).

The size distribution of droplets of the natural settling case and the ion-enhanced settling case are compared in **Figure 8**. The droplet sizes are distributed mainly in the range of $10 \sim 40$ μm for the case of natural settling. The charged aerosols are generated through the combination of ions and aerosols. Besides the conventional forces such as the thermophoretic and diffusophoretic forces, the electric forces contribute to the collisions between the small charged aerosols and uncharged aerosols. (Cherrier et al., 2017; Hashino et al., 2014; Luan et al., 2019; Roy et al., 2019; Wang and Pruppacher, 1977; Zhang et al., 2018b; Zhang and Tinsley, 2017). The charged aerosols can induce the image charge on large uncharged aerosols, and the consequent electric forces are the short range attractive force. The previous simulation results suggest that the charged aerosols can increase the collision rate between aerosols by 1 to 3 orders of magnitude(Tan et al., 2016). Therefore, the droplet sizes are distributed in the range of $50 - 80$ μm for the case of charged aerosols settling case. However, the uncharged aerosols of the natural settling case (control group) can only cause limited increase of the particle size distribution. The large droplets generated by charged droplets can capture a large number of small aerosols during their settling process and cause faster settling of supersaturated water vapor in the cloud chamber. **Figure 8 (d)** indicates that the settling amount of moisture induced by charged aerosols is 38% higher than in the natural settling case (control group) at 10 min after the start of the experiment.

**3.4 Technoeconomic estimates**

As a simple techno-economic estimate, we note that the cost of the large corona discharge system including the DC power supply, 6 poles, the wires with length of 7.2 km was less than 250,000 Chinese Yuan (CNY). The low-maintenance solar and wind power system (estimated cost of 50,000 CNY) can provide a continuous supply of renewable energy for the corona plasma discharge system. This energy supply system can be installed on-site and operated off-the-grid. Even considering the high construction cost of the system on the mid-altitude mountains to be about 200,000 CNY, the total cost of the installation was approximately 500,000 CNY. Importantly, this cost is well below the typical cost of a cloud seeding carried in China (at least 1 million CNY) to generate artificial rain. After the completion of the setup and real-world testing and optimization of the large-scale corona discharge installation, it may have an effect on the rainfall of downwind area. A similar approach can be used for fog elimination applications, for example along the airport runways. Importantly, the plasma system can be switched on to instantly generate the ions, and then switched off to also instantly cease the ion production whenever the wind direction or other atmospheric conditions change. This important feature of the plasma adds additional level of flexibility in commercial operation of the system.

3.5 Future work

We have stressed that the precipitation by charged particles actually depends on the relations between temperature, humidity supersaturation and ion concentration. The more indoor experiments within larger temperature range and humidity range can provide more detailed data to determine the relations above. The future outdoor experiment on the high mountains will determine the effect of wind, temperature and terrain on the ion coverage and precipitation range. Although the wire icing is a challenge for the outdoor experiment, the reliable ice melting system can solve this problem.

**4. Conclusion and implications**

To summarize, a large-scale corona plasma discharge system was installed to analyze the production

and the coverage area of negative ions that are capable to induce precipitation of atmospheric aerosols in downwind direction. The nitrogen species dominated the optical emission spectra of the negative corona discharges. The corona discharge was found to perform as a stable ion source with the density of $\sim 10^8/cm^3$. The coverage area of the ions was dramatically improved by using over 300,000 corona discharge points, which also reduced the common destructive interference leading to the decay of ion concentrations in the open air, thereby dramatically increasing the outward ion transport capacity of the large-scale corona plasma discharge installation. As a result, the large ion coverage area (30 m×23 m×90 m) has been achieved experimentally and validated by the numerical calculations. The faster wind speed was a more efficient way to increase the ion coverage area. The cloud chamber experiment confirms that the charged aerosols generated by ions can accelerate the settlement of moisture by 38%. These results indicate that the large scale corona discharge installation indeed can increase the ion density within a certain region. Furthermore, the ion-induced charged aerosols may realistically trigger water precipitation or, alternatively, fog elimination. Since the latter effects were studied using our large-scale cloud chamber laboratory system, systematic field studies under real-world conditions are warranted to optimize the complex processes involved in ion-induced precipitation of atmospheric aerosols under prevailing weather conditions.

*Data availability.* Data are available from the corresponding author upon a reasonable request.

*Author contributions.* SM and HC made equal contributions. DL initiated and supervised the study. JL and MX performed experimental studies. DL and KO collaborated on interpreting the results. DL prepared the manuscript, with significant contributions from HC, JL, and KO.

*Competing interests.* The authors declare that they have no conflict of interest.

*Acknowledgements.* This work was supported by the National Key Research and Development Plan of China (Grant No.2016YFC0401001)

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

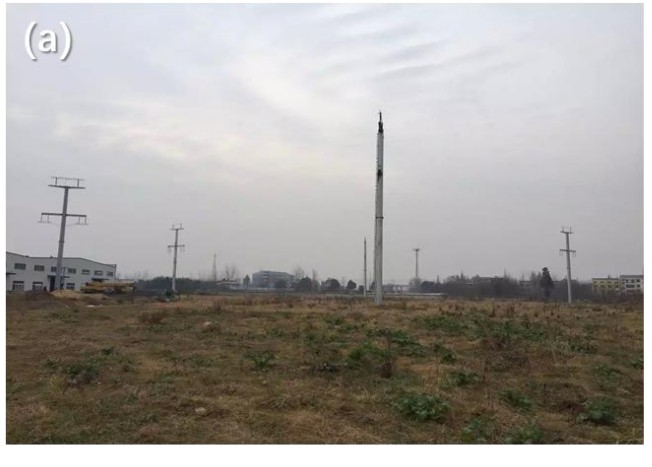
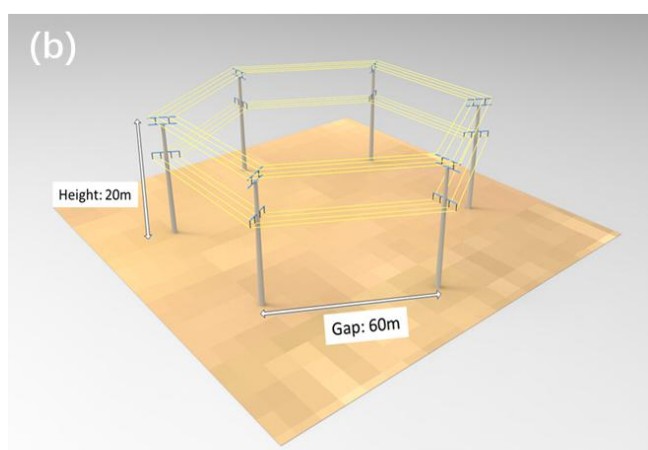
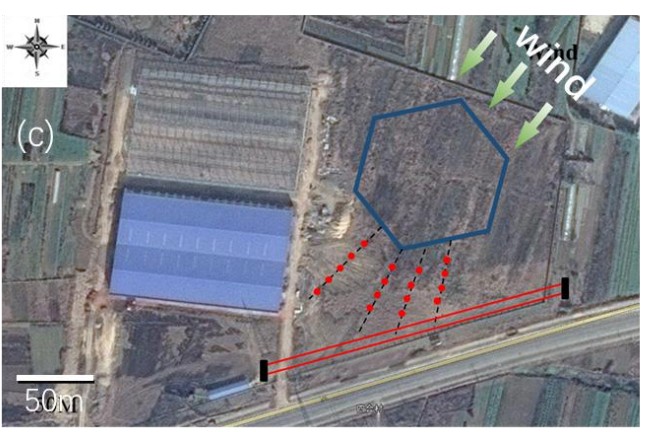
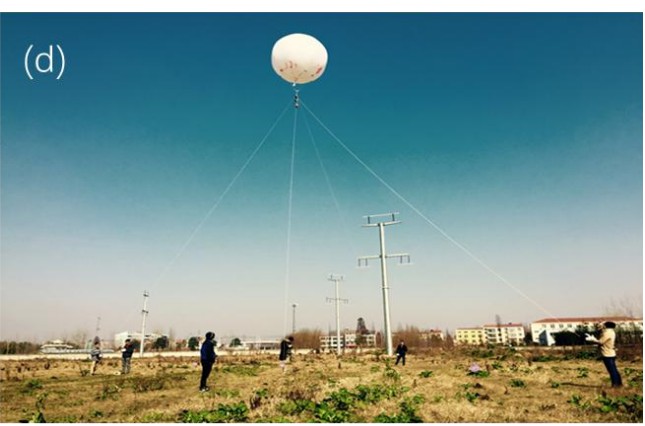

**Figure 1.** Large-scale installation for ion generation for precipitation of atmospheric aerosols, using over 300,000 corona plasma discharges. (a) The large-scale corona discharge system consists of 6 poles and 7.2 km electric wires. (b) The design diagram of the hexagonal discharge arrays. (c) The satellite image of the test zone (taken from https://map.baidu.com, last access: 29 December 2019). The hexagon shows the corona plasma discharge system. The red points show the locations of ion density measurements. (d) The hydrogen balloon carried the ion density measurement device.

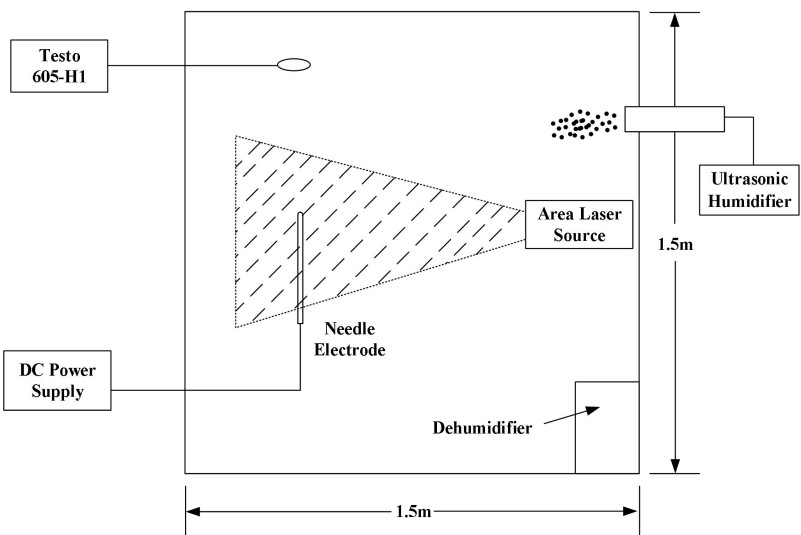

3      **Figure 2.** Schematic of the 6.75 m³ cloud chamber used for the ion-enhanced precipitation.

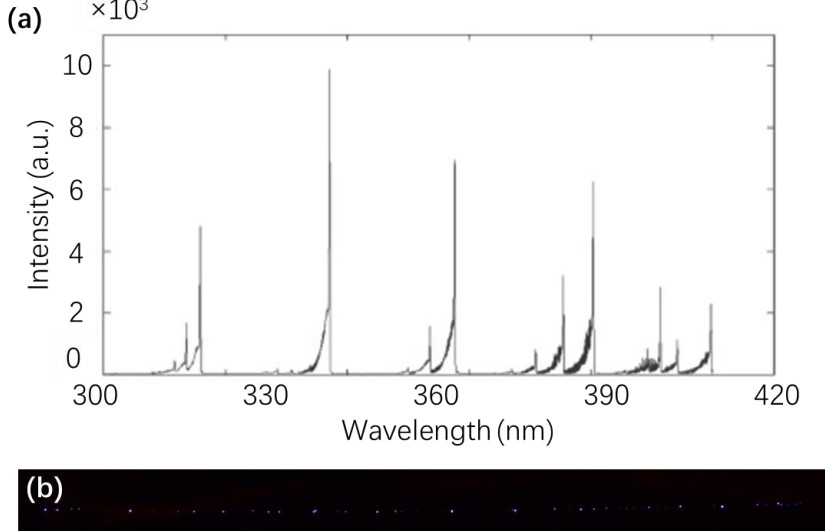

7      **Figure 3.** (a) Nitrogen species dominate the optical emission spectra (shown between 300 nm and

8      400 nm here) from the negative corona discharge on the wire electrode generated with the applied

9      voltage of -40 kV. (b) The image of corona discharge on the wire (1 m length, -40 kV). The exposure

10     time of the image is 0.5 seconds. The wire (diameter of 1 mm) was six strands of stranded wire.

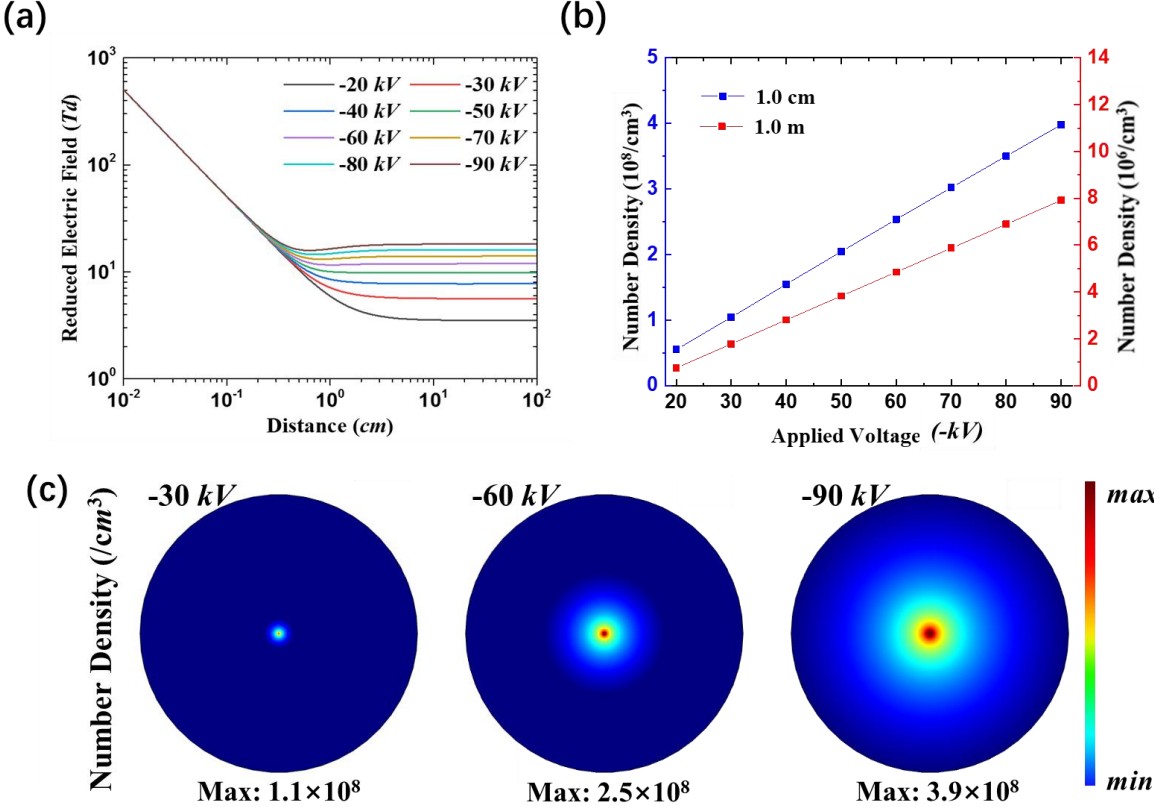

**Figure 4.** The increasing applied voltage enhances the ion output capacity of the corona discharge source. (a) The electric field from the center as a function of applied voltage.(b) The ion density at 1 cm (blue line) and 1 m (red line) from the center as a function of applied voltage. (c) The distribution of ions generated by single corona discharge point (simulation). The center is the corona discharge point. The radius of the big blue circle is 1 m.

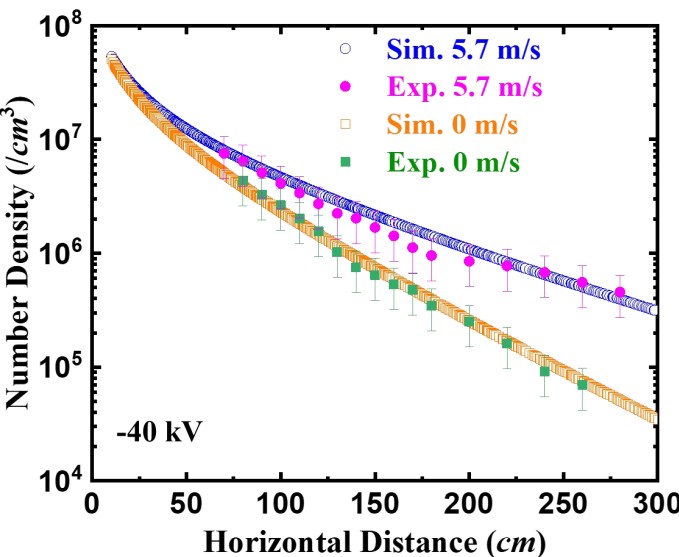

3    **Figure 5.** The effect of wind on the distribution of ions generated by a single corona discharge point

4    (indoor tests). Wind speed at 0 m/s and 5.7 m/s. The applied voltage on the wire electrode was -40

5    kV. The experiment measurement data is plotted with error bars. The simulation data is plotted by

6    hollow symbols.

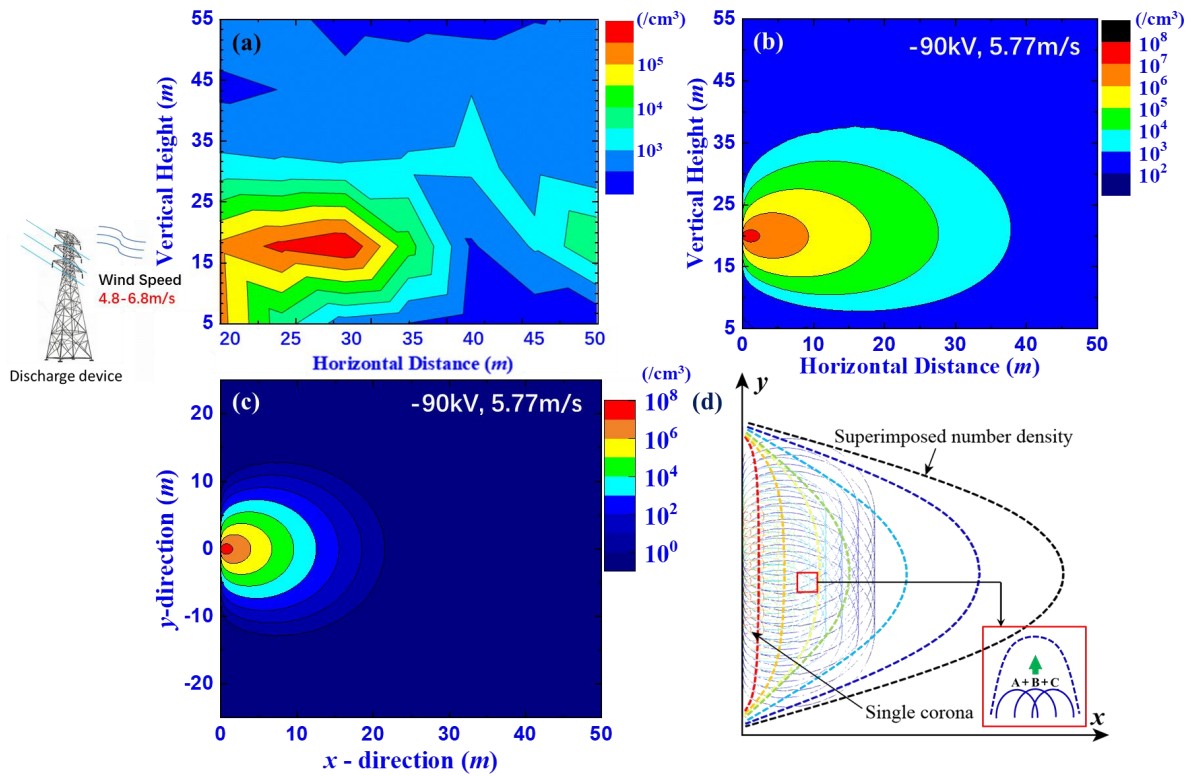

**Figure 6.** The coverage area of the large corona discharge system is the combination of the coverage areas of the very large number of corona discharge points. (a) The ion density distribution in vertical direction and downwind direction (experimental measurement, applied voltage -90 kV). (b) The ion density distribution in vertical direction and downwind direction (simulation, z axis (vertical direction), x axis (horizontal direction)). (c) The ion density distribution of single corona discharge point (-90 kV) in horizontal direction (x-y axis). (d) The combination of ions generated by multi-corona discharge points resulted in high negative ion density in the open air. The dashed color clines from red color to black color suggests the ion density decreases as the distance from the wire electrode increases. The inserted picture indicates the combination of ions generated by three corona discharge sources increases the coverage of ions substantially.

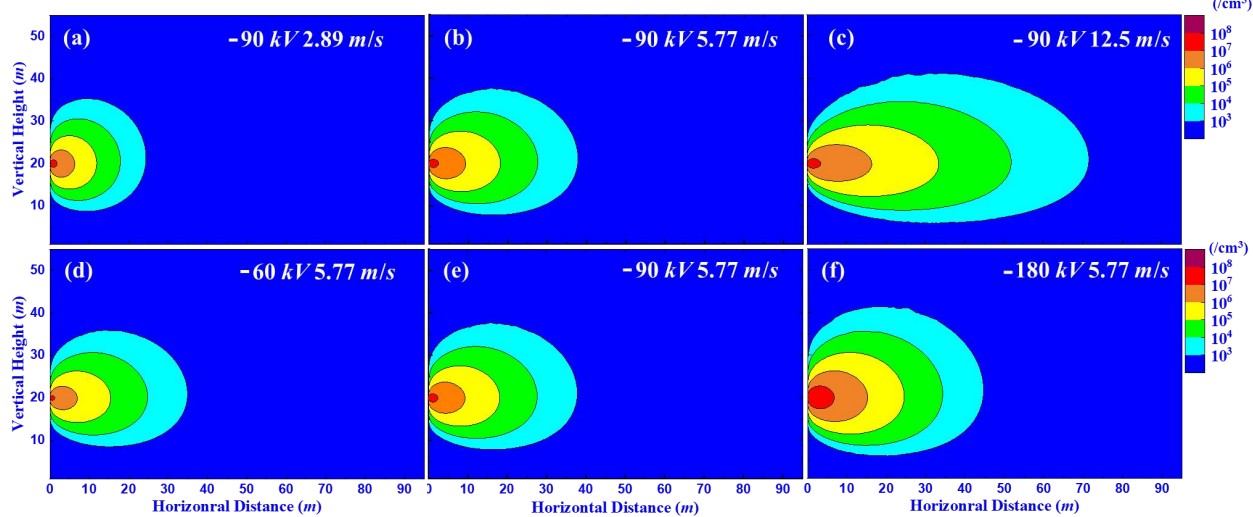

**Figure 7.** The increasing applied voltage and wind speed can enlarge the coverage area of the ion source. The effect of wind on the ion distribution in the field (numerical results). (a) -90kV, 2.89m/s (wind speed), (b)-90kV, 5.77m/s, (c)-90 kV, 12.5 m/s. The effect of wind on the ion distribution in the field (numerical results). (d) -60 kV. (e) -90 kV. (f) -180 kV.

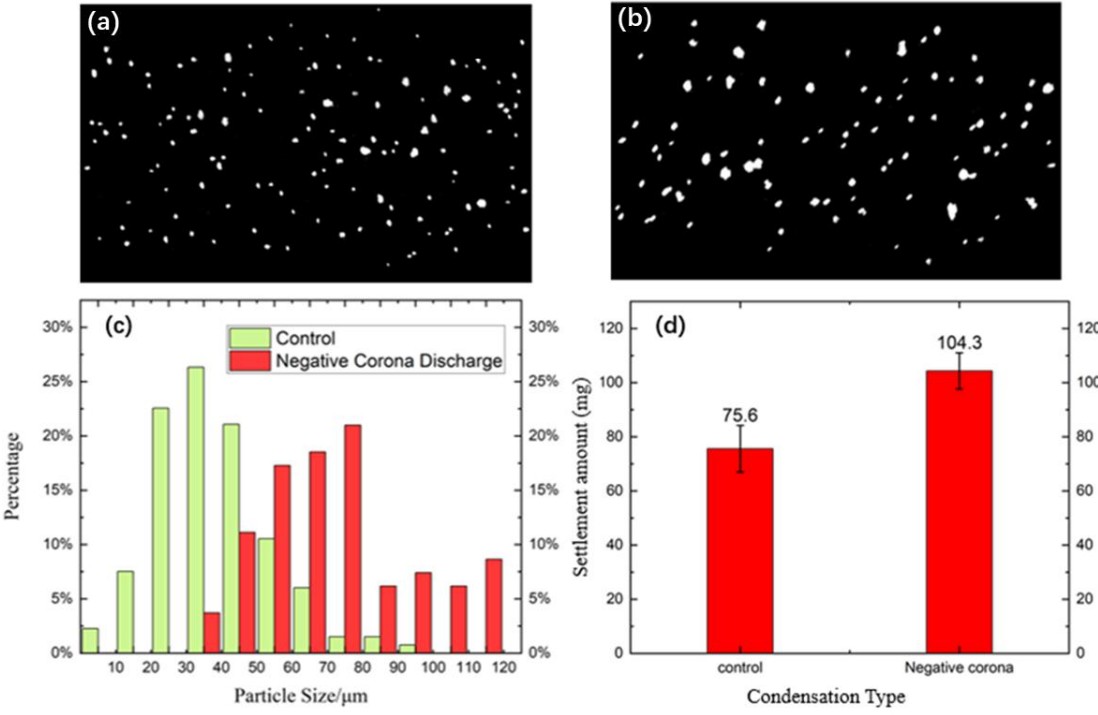

**Figure 8.** The ions generated by corona discharge source enhance the condensation of aerosols. The

photo of aerosols in the light path of a laser for the case of (a) control group, (b) the enhanced

condensation group by ions. The photo was taken 5 min after the experiment started. (c)The particle

size distribution of aerosols in the cloud chamber for the control group and negative corona discharge

group (calculated according to (a) and (b)). (d) The ions generated by corona discharge source

enhance settlement of moisture. The settlement of moisture of control, negative corona discharge

groups on the acceptor 10 min after the experiment started.

