# Peer review of "Large-scale ion generation for precipitation of atmospheric aerosols"

_Atmospheric Chemistry and Physics, 2020_

## Short Comment (SC1) · 29 Apr 2020

Figure 1(e) only shows 45 corona discharge points on the 1m long wire electrode. How did author estimate the large corona discharge system has 300,000 discharge points?

[Figure]

---

## Short Comment (SC2) · 2 May 2020

It is well known that plasma include many charged and reactive species, can author provide more details about the plasma model employed in this study? By the way, does the author has the idea to measure the charge number of the charged droplets at the humidity >100% condition?

---

## Referee Comment (RC1) · Liang Zhang (Referee) · 8 May 2020

General comments

In this paper, a large-scale device was installed in the open air, ions generation and air transport were measured; A 2-D model was developed which shown that ion concentration is pretty sensitive to wind speeds; Cloud chamber experiment further proved that ions can enhance the growth of droplets, implying a potential application of the device for rain enhancement. On the whole, this work was well designed, a heavy workload was completed, and the results are interesting and convincing. It is definitely in scope for the journal of ACP, and a decent article would be present if more details of the model and the cloud chamber experiment are given. However, frankly speaking, I don't like the writing style of this paper, and the author seems not realize that readers with atmospheric background may not be familiar with the details of plasma science. Overall, I recommend that the paper should be published once the following comments have been addressed.
* * *
Specific comments:

About the Introduction :

A paragraph should been added to describe the relevant plasma knowledge: why negative DC rather than positive DC corona discharge is selected ? the effects of the radius and material of wire? the definition of reduced electric field and source density? the species of negative ions ($O_2$?)? the mechanism of the decay of ions during transport?

About the Field experiment:

Fig. 1(e) may be move into Fig. 3.

What is the relation between the number of corona discharge points and applied voltage? I think the relevant figure should also been shown in Fig. 3.

It seems that the figures in Fig. 4 should be plot in the rank of c-b-a rather than a-b-c.

Change the title of section 3.1 to "Corona discharges and ion generation", and move the context in Line 2-13, Page 7 into section 3.1.

Fig. 6: The inset of "Discharge device" seems to be not necessary, I suggest deleting it.
Clearly plot the applied voltage and wind speed inside the Fig. 6(a)(b)(c) as Fig. 7.

The hydrogen balloon drifts between 20 m and 50 m away from the wire electrode for safety (Page 5, Line 6), how was the ion concentration within 20 m in Fig. 6(a) measured?

The source density in the model locates at the plasma boundary, just 1 cm from the wire, thus the peak in Fig. 6(b)(c) is very close to the wire. However, the peak of ion concentration in Fig. 6(a) is about 10 m ~ 20 m away from the wire, please explain why that happens.

**About the 2-D model:**
The x(downwind direction), y(wire direction), z(vertical direction) should be clearly defined in the context, which will be frequently used in Fig. 4, 5, 6 ,7.

The expression of coefficients $K$ and $\lambda$ should be given explicitly, and explain the physical processes, especially for the decay constant.

Line 23 Page 8, the decay constant $\lambda$ was reduced by 5.533 times from one single discharge point, what is the physical basis?
If the values of $\lambda$ in Fig. 7 for voltage of -60, -90, -180 kV were all reduced by 5.533 times, in consideration of the number of discharge points will change with increasing voltage?

When multiple discharge points are involved, strictly speaking, the model is no longer two-dimensional, thus more details should be given for the model in this scenario. I doubts if it is valid to build the multiple points discharge model by simply reducing the decay constant in the 2-D single point discharge model.

Will the simulated results in Fig. 7 also work at 4000 m level, considering the low pressure/density there?

**About Cloud chamber experiments:**
Line 14-15, Page 9, the ion density of $1.2*10^5$ ~$2*10^4$/cm$^3$ was provided, however, according to Fig. 4(b) and Fig. 5, the ion density at 1 m away for -23 kV should be about $1*10^6$/cm$^3$ for a single point, why the ion concentration in the chamber is so low?

What is the distribution of charges on varying size of aerosols? Is it possible to provide the average charges on aerosols through dividing the amount of charges by the amount of aerosols?

Fig. 8(d) seems not necessary.
The results will be more interesting if figures like Fig. 8(c) are also obtained at times such as 1 min, 2 min, 10 min, as they will be helpful to illustrate the mechanism through which the charged aerosols enhance the growth of droplets. If possible, show them in Fig. 8, and move Fig. 8(a)(b) to Fig. 2.

The temperature in the chamber should be provided in section 3.3.

According to Fig. 6&7, the ion concentration reduced to ~$10^3$/cm$^3$ at 50 m away, will the effect of charges on precipitation still be significant at that low concentration? If possible, provide the minimum and maximum ion concentration that could affect precipitation.
* * *
**Technical corrections:**

Page 5, Line 6: in the downwind -> in the downwind, as shown in Fig. 1(d)

Page 5, Line 2: $\nabla$is -> $\nabla$ is

Page 6, Line 3: during the experiment -> during the experiment (Testo 605-H1)

Page 7, Line 5: the reduced electric field -> the reduced electric field (electric field divided by neutral density, E/N)

Page 7, Line 5: 80 Td -> 80 Td (1 Td = ⋯ V m$^2$ )

Page 8, Line 10: -40 kV -> -90 kV

Page 8, Line 13-14: "The mutual ⋯ the wires" has already appeared in section 2.1. Delete it.

Page 9, Line 14-15: the ion density of $1.2*10^5$ ~$2*10^4$/cm$^3$ -> ???

Page 9, Line 15:   1-20 m -> ??? (20-30m?)

Page 9, Line 17: diameter > 0.7 -> diameter > 0.7 μm? Give a reference for this value.

Page 9, Line 18: contribute -> contributes

Page 9, Line 19: charging.(Jidenko⋯) -> charging (Jidenko⋯).

Page 9, Line 26: the charged aerosols -> the small charged aerosols

Page 10, Line 1: on uncharged aerosols -> on large uncharged droplets

Page 10, Line 1: "the consequent electric forces are short range attractive forces" -> the consequent image electric force is short-range attractive force.

Page 10, Line 3: magnitude.(Tan⋯) -> magnitude (Tan⋯).

Page 17, 6.75m$^3$ -> 6.75 m$^3$

Page 18, Line 6: 1 cm -> 1 cm (blue line) and 1 m (red line).

Page 19, Line 5: error bar -> error bars

Page 20, Line 8: color clines -> color lines.

Page 21, Line 5: the effect of wind on -> the effect of voltage on.

Fig. 1(a)(b)(c): plot "(a), (b), wind, 50 M" in white color; Fig. 1(c): explain the two red lines, and plot the direction chart at the lower left corner and "50 M" at the lower right corner.

Fig. 4(a), explain the meaning of '1s' in the caption.

Fig. 5: plot "-40 kV" inside the figure.

Fig. 6(a): plot "(a)" in white; Fig. 6(a)(b)(c): the legends should be better plot; Fig. 6(d): exchange the position of "y" and "(d)", move "x" below the arrow.

Fig. 8: move "(a)(b)" to the top left corner.

---

## Short Comment (SC3) · 8 May 2020

The high voltage experiment is always dangerous no matter indoor or outdoor testing. Can authors provide more details on the measuremt of ions generated by large corona discharge system? What is the relation between indoor single corona discharge source and outdoor large corona discharge system?

---

## Short Comment (SC4) · 8 May 2020

The author provide an interesting way to enhance precipitation by large corona discharge. Why are you going to setup this system on the high mountains with altidude of 3000∼4000m instead of desert?

---

## Author Comment (AC1) · 8 May 2020

SC1: 'Figure 1(e) only shows 45 corona discharge points on the 1m long wire electrode. How did author estimate the large corona discharge system has 300,000 discharge points?',

AnswerïijŽFigure 1 (e) shows there are more than 45 corona discharge points along the 1 m long wire electrode when the applied voltage is -40 kV, and the average distance between corona discharge points is only 2.5 cm. Actually, for our large corona discharge system with the 7.2 km long wire electrode and applied voltage of -90 kV, the number of the corona discharge points is expected to be at least $3.2 \times 10^5$ (45 1/m $\times 7.2 \times 10^3$m). The mutual interference between wire electrodes is also avoided by the

horizontal distance of 0.5m and the vertical distance of 5 m between the wires.

---

## Author Comment (AC2) · 8 May 2020

It is well known that plasma include many charged and reactive species, can author provide more details about the plasma model employed in this study? By the way, does the author has the idea to measure the charge number of the charged droplets at the humidity >100% condition?

Answer: The plasma model employed in this study is based on the air plasma model we used in the previous 11 papers. The model includes $O_2$, $N_2$ and $H_2O$ species, and more than 200 reactions between these species. The charged species generated by plasma in the model includes electrons, $O_2^-$, $O_2^+$, $O^+$, $N_2^+$, $H^+$, $H_2O^+$ and et al.

It is difficult to mesure the charge number of charged aerosols. However, we designed

an experiment to measure the charge amount of charged aerosols by stop the horizontal movement of charged aerosols by static electric field. We can calculate the charge amount by the horizontal speed of charged aerosols, the travelling distance of charged aerosols and electric field intensity. The paper is going to be published in Journal of physics D: Applied physics.

---

## Short Comment (SC5) · 10 May 2020

This paper provides an interesting study on the new application of large corona discharge system. What are the obstacles for the running of this huge corona discharge system on the high mountain peaks? and how to keep the discharge system operating stably?
* * *

---

## Author Comment (AC3) · 21 May 2020

The high voltage experiment is indeed dangerous. In order to ensure the experiment safety, the hydrogen balloon carrying the ion counter (Air Ion Counter) is used to measure the vertical (5 m – 50 m) and horizontal (20 m – 50 m) ion density in the downwind. The volume of hydrogen balloon is about 5 m^3. 4 people collaborate to control the movement of the hydrogen balloon. The horizontal distance of 20 m is the minimum distance from the wire electrode according to the manual for high voltage operation.

Although the indoor experiment has the lower applied voltage and shorter wire electrode, the ion density measurement at the same wind speed as the outdoor (generated by electric fan) provides data base to setup the ion transportation model. The higher

applied voltage of the outdoor experiment can generate more intense corona discharge and much more discharge points on the wire electrode. By improving the ion density at the source point and considering the combination of corona discharge points, the ion transportation model can also apply to the outdoor experiment.
* * *

---

## Author Comment (AC4) · 21 May 2020

The high ion density generated by large corona discharge system can enhance precipitation at the condition of humidity supersaturation. The plan is to setup the large corona discharg system on the windward slopes near the mountain peak, so the high ion density can play a fuller role in the condition of humidity supersaturation and enhance precipitation in the downwind region. If the large corona discharge system is setup in the dezert, it is difficult for ions to rise towards cloud during their lifetime.

---

## Author Comment (AC5) · 21 May 2020

The first challenge is to provide stable electric power to the discharge system. The combined system of wind and solar power system designed by our electric power team can provide enough power to run the discharge system. However, we need build a stable room with strong air drier to alleviate the side effect of super-humidity on the high voltage DC power supply. The second issue is that the system has to be shut down when temperature < 0 degree at the moment, because the wire electrode is easily frozen if it works at this low temperature condition. In order to solve this problem, an ice-melting system for the wire electrode is being tested now. We can keep the discharge system run properly by solving these two issues.

---

## Author Comment (AC6) · 25 May 2020

Specific comments:

**About the Introduction :**

A paragraph should been added to describe the relevant plasma knowledge: why negative DC rather than positive DC corona discharge is selected ? the effects of the radius and material of wire? the definition of reduced electric field and source density? the species of negative ions ($O_2$?)? the mechanism of the decay of ions during transport?

Answer:

The content of answers to the question above is more than one paragraph. To put them in the section 2.1 and explain why we choose corona discharge source is more appropriate.

Both the positive and negative corona discharge can be used to increase the ion density in the open air. Under the similar conditions of the electric circuit the loss of the positive corona is greater than that of negative corona at the same applied voltage. Because the negative corona curve is flatter and since larger negative corona currents can be obtained, the negative corona is much better adapted for the application such as fog elimination and electrical precipitation than the positive corona.(Sawant et al., 2012; Strong, 1913)

The wire electrode is a low cost and high efficiency plasma source configuration, especially for the large scale corona discharge system. For the wire electrode radius within the range of 100 μm to 1000 μm, the plasma thickness increases with increasing wire radius. The larger wires can generate more electrons, however, the electron energy decreases due to the lower electric field near the larger wire(Chen and Davidson, 2003). Stainless steel stranded wire is suitable wire electrode material considering durability and stability.

Electrons generated by negative corona discharge attach to electronegative gas molecules (such as, $O_2$) to generate negative ions ($O_2^-$). Recombination of electrons with positive ions is negligible. Therefore, ionization competes primarily with electron attachment. The ionization predominates over the electron attachment and new electrons are generated. The rate of ionization balances the rate of electron attachment at the reduced electric field of 120 Td (1 Td $=10^{-21}$ Vm$^2$). Beyond this ionization boundary, the attachment dominates over the ionization, and the electron density decrease gradually as the electric field decreases.(Chen and Davidson, 2003; Kossyi et al., 1992; Lowke and Morrow, 1994) In the region away from the electrode, because the absence of the electric field, the charged particles, including electrons and ions, perform a faster decay through the electron-ion and ion-ion recombination with background charged particles.(Xiong et al., 2010)

**About the Field experiment:**

Fig. 1(e) may be move into Fig. 3.

What is the relation between the number of corona discharge points and applied voltage? I think the relevant figure should also been shown in Fig. 3.

Answer: Figure 1 (e) is changed to Figure 3 (b) as the figure below

[Figure]

The increasing applied voltage increased the plasma power indeed. Our experiment results also indicated that as the applied voltage increased from -30kV to -40kV, besides the corona discharge points increased by ~10%, the OES intensity increased by ~20%.

It seems that the figures in Fig. 4 should be plot in the rank of c-b-a rather than a-b-c.

**Answer**: Figure 4 is replotted by the order of "c-b-a" as the figure below.

[Figure]

Change the title of section 3.1 to "Corona discharges and ion generation", and move the context in Line 2-13, Page 7 into section 3.1.

**Answer**: The title of section 3.1 is changed to "Corona discharges and ion generation". The context

in Page 7 is moved into section 3.1.

Fig. 6: The inset of "Discharge device" seems to be not necessary, I suggest deleting it.

Clearly plot the applied voltage and wind speed inside the Fig. 6(a)(b)(c) as Fig. 7.

The hydrogen balloon drifts between 20 m and 50 m away from the wire electrode for safety (Page 5, Line 6), how was the ion concentration within 20 m in Fig. 6(a) measured?

The source density in the model locates at the plasma boundary, just 1 cm from the wire, thus the peak in Fig. 6(b)(c) is very close to the wire. However, the peak of ion concentration in Fig. 6(a) is about 10 m ~ 20 m away from the wire, please explain why that happens.

**Answer**: Because the hydrogen balloon drifts between 20 m and 50 m and we did not measure the ion density within 20m, the x-axis of Figure 6 (a) should be 20m ~ 50m. The previous x-axis label of 0-50m is a mistake. The applied voltage and wind speed are also added on the Figure 6 (b) and (c).

The peak concentration at the position between 25m and 30m in Figure 6 (a) is $5.6*10^6/cm^3$, and the concentration in the close region is about $3*10^6/cm^3$. The color scale of Origin (software to plot the figure) make the peak concentration is more visible. The reason of this peak is the randomness of wind. The overall trend of Figure 6 (a) is the decreasing ion density as the distance increases, which is consistent with the simulation results of Fig. 6(b)(c). Figure 6 is replotted as the figure below.

[Figure]

**About the 2-D model:**

The x(downwind direction), y(wire direction), z(vertical direction) should be clearly defined in the context, which will be frequently used in Fig. 4, 5, 6 ,7.

The expression of coefficients K and $\lambda\lambda$ should be given explicitly, and explain the physical processes, especially for the decay constant.

**Answer**: The explain of x (horizontal downwind direction), y(horizontal radial direction), z(vertical direction) is added.

The eddy diffusion $K$ represents the diffusion of ions under the influence of the turbulent state of atmosphere. During stable conditions, the maximum value of eddy diffusivity decreases with increasing stability. In stable conditions, a height at which turbulence maintained is limited by the destruction of turbulent kinetic energy by negative buoyancy(Ulke, 2000), while in unstable conditions, the maximum value of eddy diffusivity increases with growing instability characterized by increasing values of $H_A/L$ ($H_A$ is the ABL-height, L is the Monin-Obukhov length).

The decay constant $\lambda$, it represents the decay of ions due to the recombination reactions between charged particles, such as $e+N_2^+ \rightarrow N_2$, $e+O_2^+ \rightarrow 2O$, $O_2^- + N_2^+ + N_2 \rightarrow 2O + 2N_2$, etc. According to our simulation results, the combination of numerous corona discharge points actually decreases the decay of ions generated by a single corona discharge point in the open air.

Line 23 Page 8, the decay constant $\lambda\lambda$ was reduced by 4.533 times from one single discharge point, what is the physical basis?

**Answer**: The prototype of our model is designed to simulate the transportation and decay of ions generated by one source point, but the large scale corona discharge system consists of many corona discharge source points, in order to simulate the enhanced transportation effect of multi corona discharge points, decreasing the decay constant of the model by 4.533 can get the similar result as the experiment measurement.

If the values of $\lambda\lambda$ in Fig. 7 for voltage of -60, -90, -180 kV were all reduced by 4.533 times, in consideration of the number of discharge points will change with increasing voltage?

**Answer**: To be honest, because our large scale corona discharge experiment was only carried out with the applied voltage of -90 kV, the $\lambda$ decreased by 4.533 was used for all simulation cases after the checking between the simulation and experiment measurement case at -90 kV. The further experiment will be carried out to get more accurate $\lambda$ for the cases of -60, -90, -180 kV.

When multiple discharge points are involved, strictly speaking, the model is no longer two-dimensional, thus more details should be given for the model in this scenario. I doubts if it is valid to build the multiple points discharge model by simply reducing the decay constant in the 2-D single point discharge model. Will the simulated results in Fig. 7 also work at 4000 m level, considering the low pressure/density there?

**Answer**: We absolutely agreed with editor's suggestions. To setup the accurate the transportation model for multi corona discharge points need consider the interactions between corona discharge points, wind direction and speed, temperature, pressure, background ion density, and decay reactions of ions in the air. The model at the moment is only a prediction model. Once the construction of large scale corona discharge system on the high mountain is finished, the accurate measurement can provide the essential data for the model.

**About Cloud chamber experiments:**

Line 14-15, Page 9, the ion density of $1.2*10^5 \sim 2*10^4/cm^3$ was provided, however, according to Fig. 4(b) and Fig. 5, the ion density at 1 m away for -23 kV should be about $1*10^6/cm^3$ for a single point, why the ion concentration in the chamber is so low?

**Answer**: Fig. 4(b) and Fig. 5 show the ion density of the corona discharge on 1m long wire electrode.

The chamber experiment want to prove the ion density of the measurement region can enhance the precipitation of droplet, therefore, a single needle electrode is used to generate a single corona discharge point and provide the ion density at the position of 30m-35m from the large scale corona discharge system in the chamber. The higher ion density can be obtained by increasing the applied voltage or using the wire electrode.

What is the distribution of charges on varying size of aerosols? Is it possible to provide the average charges on aerosols through dividing the amount of charges by the amount of aerosols?

**Answer**: We have measured the charge amount of charged aerosols. The idea is to let aerosols pass through the plasma generated by plasma jet, and then use the static electric field to decelerate the moving charged aerosols, which is just like "brake experiment". Because the mass, velocity and the travelling distance of charged aerosols, and electric field intensity are known, we can calculate the charge amount of charged aerosols. This paper is going to be published in "Plasma Science and Technology". This method is suitable for highly charged aerosols generated by plasma jet, although it's more accurate, we are still working to improve it to be suitable for low charged aerosols generated by the corona discharges.

Fig. 8(d) seems not necessary.
The results will be more interesting if figures like Fig. 8(c) are also obtained at times such as 1 min, 2 min, 10 min, as they will be helpful to illustrate the mechanism through which the charged aerosols enhance the growth of droplets. If possible, show them in Fig. 8, and move Fig. 8(a)(b) to Fig. 2.
The temperature in the chamber should be provided in section 3.3.

**Answer**: According to our previous experiment experience, the difference between charged and uncharged aerosols are negligible at 1 min, and there are much less charged aerosols than the uncharged aerosols at 10 min. Because of COVID-19, our university (Huazhong University of Science and Technology, Wuhan) is closed and every lab is shut down at the moment. We will probably start new experiment in September. So, we can not plot the condition at 1 min, 2min and 10 min now. The temperature inside the chamber is $2\pm1°C$, which is provided in the section 3.3.

According to Fig. 6&7, the ion concentration reduced to $\sim10^3/cm^3$ at 50 m away, will the effect of charges on precipitation still be significant at that low concentration? If possible, provide the minimum and maximum ion concentration that could affect precipitation.

**Answer**: The minimum ion concentration can induce precipitation of droplet is about $10^5/cm^3$ with the humidity supersaturation at $130\pm10\%$. We tried the ion concentration less than $5\times10^4/cm^3$, but there was no significant effect. The higher ion concentration, such as $10^6/cm^3$ and even higher, obviously can accelerate the precipitation. The precipitation of droplets by ions actually depends on the relations between temperature, humidity supersaturation and ion concentration. To analyze these principles is our next work.

Technical corrections:
**Answer**: All these mistakes have been corrected.

Page 5, Line 6: in the downwind -> in the downwind, as shown in Fig. 1(d)

Page 5, Line 2: $\nabla$is -> $\nabla$ is

Page 6, Line 3: during the experiment -> during the experiment (Testo 605-H1)

Page 7, Line 5: the reduced electric field -> the reduced electric field (electric field divided by neutral density, E/N)

Page 7, Line 5: 80 Td -> 80 Td (1 Td = … V $m_2$ )

Page 8, Line 10: -40 kV -> -90 kV

Page 8, Line 13-14: "The mutual … the wires" has already appeared in section 2.1. Delete it.

Page 9, Line 14-15: the ion density of $1.2*10_5 \sim 2*10_4/cm_3$ -> ???

Page 9, Line 15: 1-20 m -> ??? (20-30m?)

Page 9, Line 17: diameter > 0.7 -> diameter > 0.7 μm? Give a reference for this value.

Page 9, Line 18: contribute -> contributes

Page 9, Line 19: charging.(Jidenko…) -> charging (Jidenko…).

Page 9, Line 26: the charged aerosols -> the small charged aerosols

Page 10, Line 1: on uncharged aerosols -> on large uncharged droplets

Page 10, Line 1: "the consequent electric forces are short range attractive forces" -> the consequent image electric force is short-range attractive force.

Page 10, Line 3: magnitude.(Tan…) -> magnitude (Tan…).

Page 17, $6.75m_3$ -> 6.75 $m_3$

Page 18, Line 6: 1 cm -> 1 cm (blue line) and 1 m (red line).

Page 19, Line 5: error bar -> error bars

Page 20, Line 8: color clines -> color lines.

Page 21, Line 5: the effect of wind on -> the effect of voltage on.

Fig. 1(a)(b)(c): plot "(a), (b), wind, 50 M" in white color; Fig. 1(c): explain the two red lines, and plot the direction chart at the lower left corner and "50 M" at the lower right corner.

Fig. 4(a), explain the meaning of '1s' in the caption.

Fig. 5: plot "-40 kV" inside the figure.

Fig. 6(a): plot "(a)" in white; Fig. 6(a)(b)(c): the legends should be better plot; Fig. 6(d): exchange the position of "y" and "(d)", move "x" below the arrow.

Fig. 8: move "(a)(b)" to the top left corner.

---

## Referee Comment (RC2) · Anonymous Referee #1 · 24 Jun 2020

Overall, the author's answers in the open discussion are reasonable and specific. The paper can be published in the current version. The author should pay attention to make the corresponding corrections in the final version of the paper.

---

## Referee Report (RR1)

My previous comments have been adequately addressed in the author's response, I think the revised manuscript is more convincing, and it should been accepted after minor revisions with respect to the following suggestions:

1. The eddy diffusion K and decay constant $\lambda$ were described with more details, however, it should be explicitly shown if they are constants or they vary with factors such as voltage, wind speed, humidity, temperature and pressure.

2. The steady-2D equation (1) seems not complex, and the results in Fig. 4 and Fig. 7 seems regular, thus I guess there may exist an analytic solution of ion density for varying wind speeds and voltages, at least along the x-axis. Is it possible to reach such a result in the further?

3. In comparison with the single-discharge-point results of simulation and indoor experiment in Fig. 5, I prefer to see the relevant multi-discharge-point results, which is more realistic.

4. In page 9: "the whole coverage volume was approximately 30m*20m*90m". Firstly, how the width of 90m was obtained? as in Fig. 1(b) the distance between two poles is only 60m. Secondly, according to Fig. 6(d), the superimposed ion density decays at the boundary, what is the length of boundary? should it be taken away from the width of 90m?

5. Although the coverage of ion density in Fig. 6(b) is coincident with that in Fig. 6(a), the ion density in Fig. 6(a) ranges between $10^6$ and $10^5$ at the distance of 20m~30m, while in Fig. 6(b) it is less with about one order of magnitude, ranging between $10^5$ and $10^4$. Please discuss where the difference results from and how to improve it.
Besides, the x-ticks in Fig. 6(a) seems to be not in line with the x-names.

6. The results in Fig. 8 support that the ion density in the region of 30m-35m in Fig. 6(a) contributes to precipitation, however, the minimum of ion density which can enhance settling should be obtained in order to estimate the largest effective distance in Fig. 6(a).

7. The authors should carefully check the values, units, references (a lot errors), figures (obscures seen in printout) in the paper one by one, to avoid unnecessary mistakes.

Some of above suggestions maybe beyond the scope of this paper, it is preferable to add a section to discuss further works to make the paper more coherent.

---

## Author Response (AR2)

My previous comments have been adequately addressed in the author's response, I think the revised manuscript is more convincing, and it should been accepted after minor revisions with respect to the following suggestions:

**Comment 1**.The eddy diffusion K and decay constant $\lambda$ were described with more details, however, it should be explicitly shown if they are constants or they vary with factors such as voltage, wind speed, humidity, temperature and pressure.

**Response:**

We greatly appreciate your comment. The eddy diffusivity $K$ is calculated though the meteorological data of the test place, including: the atmospheric boundary layer height (about 300~1000m, determined by the potential temperature profile), the frictional velocity (calculated through the scale of the roughness (roughness length), which is about 0.01-0.04 for the ground), Monin-Obukhov length (about -59.8 m calculated in the model), and wind speed (which is varied with vertical height).

The detail formula can be found in Ref [1]. The simulation results show that the eddy diffusion is larger at higher vertical height (<100 m due to the simulated geometry restriction), and the value is about 4.82 m$^2$/s at 20 m vertical height (the corresponding wind speed is 5.77m/s).

For the decay rate $\lambda$, it represents the ions decay rate due to the recombination between ions and electrons, such as e + N$_2^+$→2N, e + O$_2^+$→2O, etc. Table 1 shows the recombination rates of all reactions selected in the model. [2]

Table 1 the recombination reactions

| Reactions | Rate | Ref. |
|---|---|---|
| e+N$_2^+$+M→N$_2$+M | $3.12\times10^{-35}T_e^{-1.5}$ | 2 |
| e+N$_2^+$→N+N($^2$D) | $1.50\times10^{-12}T_e^{-0.7}$ | 2 |
| e+N$_2^+$→N+N | $1.66\times10^{-12}T_e^{-0.7}$ | 2 |
| e+N$^+$+M→N+M | $3.12\times10^{-35}T_e^{-1.5}$ | 2 |
| e+O$_2^+$+M→O$_2$+M | $3.12\times10^{-35}T_e^{-1.5}$ | 2 |
| 2e+O$_2^+$+M→e+O$_2$ | $1.00\times10^{-31}T_e^{-1.5}$ | 2 |
| e+O$_2^+$→O+O(1D) | $1.24\times10^{-11}T_{eg}^{-4.5}$ | 2 |
| e+O$_2^+$→O+O | $1.68\times10^{-11}T_e^{-0.7}$ | 2 |
| e+O$^+$+M→O+M | $3.12\times10^{-35}T_e^{-1.5}$ | 2 |
| e+H$_2$O$^+$→H+OH | $2.73\times10^{-12}T_e^{-0.5}$ | 2 |
| e+ H$_2$O$^+$→O+H$_2$ | $1.37\times10^{-12}T_e^{-0.5}$ | 2 |
| O$^-$+O$_2^+$→O+O2 | $2.00\times10^{-13}T_{eg}^{-0.5}$ | 2 |
| O$^-$+N$_2^+$→O+N2 | $2.00\times10^{-13}T_{eg}^{-0.5}$ | 2 |

where, M represents the neutral species, $T_{eg}=T_e(eV)/T_g(eV)= T_e(K)/T_g(K)$, and $T_e$ is electron temperature, $T_g$ is gas temperature.

The reaction rates shown in Table 1 are related to the electron temperature and gas temperature. Since the electron temperature is determined by the discharge voltage according to the Poisson's equation and the energy conservation equation, therefore, besides the number densities of ions, the decay rate $\lambda$ is also related to the gas temperature and the applied voltage (at 1 atm).

However, due to the tremendous complexity of reactions of air plasma, it is currently

really difficult for us to compute the decay λ precisely. Therefore, we have to simplify the decay λ to be a constant. In this paper, the fitted decay constant λ is obtained through experimental results, specifically, the value is 1.5113/s at the 90 kV voltage condition (outdoor). Precise calculation of the decay rate λ will be a subject of our future work.

[1] Albani, R. A. S., Duda, F. P. and Pimentel, L. C. G.: On the modeling of atmospheric pollutant dispersion during a diurnal cycle: A finite element study, Atmospheric Environment, 118, 19–27, 2015.
[2] Sakiyama, Y., Graves, D. B., Chang, H. W., Shimizu, T., & Morfill, G. E. (2012). Plasma chemistry model of surface microdischarge in humid air and dynamics of reactive neutral species. Journal of Physics D: Applied Physics, 45(42), 425201.

**Comment 2**. The steady-2D equation (1) seems not complex, and the results in Fig. 4 and Fig. 7 seems regular, thus I guess there may exist an analytic solution of ion density for varying wind speeds and voltages, at least along the x-axis. Is it possible to reach such a result in the further?

**Response**:

Thanks for your constructive comment. Figure. 4 is the results of corona discharge, which were obtained through plasma governing equations, including Poisson's equation, particle balance equations and energy conservation equation. The corona discharge simulation provides the initial value of ion density to equation (1).

The ion density distribution shown in Figure 7 is obtained through 2D finite element method. These results were based on an extremely fine mesh, of which the minimum element size is less than 0.001m (shown in Figure.1), to guarantee the accuracy as high as possible.

[Figure]

Figure.1 finite element mesh.

We note that the eddy diffusion and wind velocity vary with vertical height ($z$). Therefore, it is really difficult to solve the analytic solution, and we have not found any report on the analytic solution of equation (1) under relevant conditions.

For the 1d case, when the equation (1) reaches the stationary state ($\frac{\partial c}{\partial t} = 0$), the equation can be simplified as follows:

$$u \frac{\partial c}{\partial x} = K \frac{\partial^2 c}{\partial x^2} - \lambda c,$$

where $u$ and $K$ are the constants at z=20m position, therefore, the above equation is a typical second order ordinary differential equation, and therefore the analytic solution has the general form: $c_1 e^{\gamma_1 x} + c_2 e^{\gamma_2 x}, c_1 + c_2 = c_0$ and $\gamma_1$, $\gamma_2$ are the roots of the characteristic equation: $K\gamma^2 - u\gamma - \lambda = 0$ 。

Figure 2 below shows that the analytic solution is well consistent with the numerical results (1d model, -90kV, 5.77m/s), which also proves the accuracy of our 2D simulation.

[Figure]

Figure.2 1d model results.

**Comment 3**. In comparison with the single-discharge-point results of simulation and indoor experiment in Fig. 5, I prefer to see the relevant multi-discharge-point results, which is more realistic.

**Response:**
The multi-discharge-point results is more realistic indeed. The indoor high voltage experiment is very dangerous especially when the applied voltage is -40 kV. The single discharge point is chosen because of its smaller discharge power. The figure below is the result of multi-discharge-points on the 1m long wire electrode. Because of much larger discharge power, the measurement has to start from 3.5m and end at 4.5m (the limited space in our lab obviously affect the measurement result). The electric fan only 1m behind the discharge point is shut down due to static problem, so the effect of wind speed is not measured for this case.

[Figure]

**Comment 4**. In page 9: "the whole coverage volume was approximately 30m*20m*90m". Firstly, how the width of 90m was obtained? as in Fig. 1(b) the distance between two poles is only 60m. Secondly, according to Fig. 6(d), the superimposed ion density decays at the boundary, what is the length of boundary? should it be taken away from the width of 90 m?

**Response:**
The overall discharge system configuration is a hexagon with the side length of 60 m, therefore, the distance between two opposite sides is 103.92m. The horizontal measurement range is ~95m with the consideration of surrounding buildings. Therefore, the width of 90 m is obtained.

Because 90m is less than 103.92m, the decays at the boundary is avoided in this range. The coverage volume of 30m*20m*90m is therefore a relatively conservative range.

**Comment 5**. Although the coverage of ion density in Fig. 6(b) is coincident with that in Fig. 6(a), the ion density in Fig. 6(a) ranges between 10^6 and 10^5 at the distance of 20m~30m, while in Fig. 6(b) it is less with about one order of magnitude, ranging between 10^5 and 10^4. Please discuss where the difference results from and how to improve it. Besides, the x-ticks in Fig. 6(a) seems to be not in line with the x-names.

**Response:**
The measurement value in Fig.6 (a) is about 3 or 4 times higher than the calculated value shown in Fig. 6 (b). The measurement error of the ion counter and unstable wind speed are the reason for that. More measurement at low wind speed condition can provide more

accurate parameters for model correction.

The discharge device shown in Fig.6 (a) is the horizontal "0" position, and the measurement starts at 20m from the wire electrode, therefore, x-ticks in Fig. 6(a) seems to be not in line with the x-names.

**Comment 6**. The results in Fig. 8 support that the ion density in the region of 30m-35m in Fig. 6(a) contributes to precipitation, however, the minimum of ion density which can enhance settling should be obtained in order to estimate the largest effective distance in Fig. 6(a).

**Response:**

The minimum ion density to induce precipitation in the cloud chamber is $\sim 2 \times 10^4/cm^3$. We do acknowledge that our work is the relatively early step towards the exhaustive studies of the precipitation effect of charged aerosols. In part, this is because the cloud chamber provides an idealized laboratory-scale condition for the precipitation experiment of charged particles. The outdoor experiment on the large-scale corona discharge system is being tested on the high mountains. Although it has some effect, we still try to find the relation between the discharge power, ion coverage range, and precipitation range. However, currently this is really a challenge, because of the strong wind, thick fog, and frozen snow. Until the end of 2019, we lost two UAVs because of sudden appearance of fog, 1 high voltage sources burned out because of the dehumidifier out of order. The estimated minimum ion density to induce precipitation in the open air will be found when the experiment starts again after we will return back to normal working conditions after COVID-19 and its aftermath are over. It is impossible to predict when exactly this will happen.

**Comment 7**. The authors should carefully check the values, units, references (a lot errors), figures (obscures seen in printout) in the paper one by one, to avoid unnecessary mistakes.

**Response**:12 errors have been corrected.

**Comment 8**. Some of above suggestions maybe beyond the scope of this paper, it is preferable to add a section to discuss further works to make the paper more coherent.

**Response**:

We thank the reviewer for this comment and the appreciation that some of the comments are indeed beyond the scope or current physical abilities of the authors to carry further research.

As recommended, we have added a brief section named "3.5 Future work" to discuss the challenges and future work. In particular, we have stressed that the precipitation by charged particles actually depends on the relations between temperature, humidity supersaturation and ion concentration. The more indoor experiments within larger temperature range and humidity range can provide more detailed data to determine the relations above. The future outdoor experiment on the high mountains will determine the effect of wind, temperature and terrain on the ion coverage and precipitation range. Although the wire icing is a challenge for the outdoor experiment, the reliable ice melting system can solve this problem.